# Inducible miR-150 Inhibits Porcine Reproductive and Respiratory Syndrome Virus Replication by Targeting Viral Genome and Suppressor of Cytokine Signaling 1

**DOI:** 10.3390/v14071485

**Published:** 2022-07-07

**Authors:** Sihan Li, Xuan Zhang, Yao Yao, Yingqi Zhu, Xiaojie Zheng, Fang Liu, Wenhai Feng

**Affiliations:** 1State Key Laboratory of Agrobiotechnology, College of Biological Sciences, China Agricultural University, Beijing 100193, China; lisihan@cau.edu.cn (S.L.); sz20163020169@cau.edu.cn (X.Z.); yaoyaoy@cau.edu.cn (Y.Y.); sz20173020143@cau.edu.cn (Y.Z.); zhengxiaojie@cau.edu.cn (X.Z.); liufang@cau.edu.cn (F.L.); 2Frontiers Science Center for Molecular Design Breeding, College of Biological Sciences, China Agricultural University, Beijing 100193, China; 3Ministry of Agriculture Key Laboratory of Soil Microbiology, College of Biological Sciences, China Agricultural University, Beijing 100193, China; 4Department of Microbiology and Immunology, College of Biological Sciences, China Agricultural University, Beijing 100193, China

**Keywords:** HP-PRRSV, miR-150, PKC, SOCS1, IFN

## Abstract

Hosts exploit various approaches to defend against porcine reproductive and respiratory syndrome virus (PRRSV) infection. microRNAs (miRNAs) have emerged as key negative post-transcriptional regulators of gene expression and have been reported to play important roles in regulating virus infection. Here, we identified that miR-150 was differentially expressed in virus permissive and non-permissive cells. Subsequently, we demonstrated that PRRSV induced the expression of miR-150 via activating the protein kinase C (PKC)/c-Jun amino-terminal kinases (JNK)/c-Jun pathway, and overexpression of miR-150 suppressed PRRSV replication. Further analysis revealed that miR-150 not only directly targeted the PRRSV genome, but also facilitated type I IFN signaling. RNA immunoprecipitation assay demonstrated that miR-150 targeted the suppressor of cytokine signaling 1 (SOCS1), which is a negative regulator of Janus activated kinase (JAK)/signal transducer and activator of the transcription (STAT) signaling pathway. The inverse correlation between miR-150 and SOCS1 expression implies that miR-150 plays a role in regulating ISG expression. In conclusion, miR-150 expression is upregulated upon PRRSV infection. miR-150 feedback positively targets the PRRSV genome and promotes type I IFN signaling, which can be seen as a host defensive strategy.

## 1. Introduction

Porcine reproductive and respiratory syndrome (PRRS) has become an epidemic in pigs worldwide since it was first reported in the United States in 1987 [1]. PRRS is caused by porcine reproductive and respiratory syndrome virus (PRRSV), a positive single-stranded RNA virus that belongs to the *Arteriviridae* family in the order *Nidovirales*. PRRSV species are divided into type 1 (PRRSV-1) and type 2 (PRRSV-2), only sharing 50–70% nucleotide similarity in their genomes [2,3]. The PRRSV genome is approximately 15.4 kb in length and contains 11 open reading frames (ORFs), 5′-untranslated region (5′UTR), and 3′UTR [4]. The 3′UTR is critical to arterivirus replication and life cycle, which can be modulated by the interaction between 3′UTR and cellular genes [5,6,7,8]. Since the highly pathogenic PRRSV (HP-PRRSV) emerged in 2006, it has been becoming a major problem for the swine industry in China and Southeast Asia [9,10,11]. However, current strategies to control PRRS, including vaccination, have not provided us with satisfactory outcomes [12]. Thus, it is necessary for us to study new ideas such as cellular miRNAs-mediated RNA interference (RNAi) to minimize PRRS impacts on the swine industry [13].

microRNAs (miRNAs) are ~19–24 nt small non-coding RNAs, which participate in almost all life processes, including cellular differentiation, development, metabolism, and viral infection [14,15]. miRNAs regulate gene expression post-transcriptionally through degradation of mRNA or inhibition of translation. For the purpose of combating viral infection, hosts have evolved multiple mechanisms to enhance defenses [16]. As a regulator of nearly all cellular pathways, miRNAs exert multiple strategies to regulate virus infection by directly targeting the viral genome, modulating antiviral signaling pathways, or targeting other host factors involved in virus infection and replication [17,18,19,20].

The first line of host defense to virus infection is innate immune response. Upon RNA virus infection, pattern recognition receptors such as retinoic acid-inducible gene I (RIG-I)-like receptors (RLR) and the Toll-like receptors (TLRs), induce the production of type I interferons (IFNs) through triggering signaling cascades [21]. Type I IFNs that are important antiviral cytokines can induce hundreds of IFN-stimulated genes (ISGs) through the JAK/STAT signaling pathway; thus, expanding antiviral responses [22,23]. Upregulation of ISGs by subsequent IFN signaling is critical to limit virus reproduction [24]. Up to now, multiple miRNAs have been revealed to regulate the production of type I IFN and ISGs expression; thus, modulating PRRSV replication. miR-30c, miR-382-5p, and miR-373 are demonstrated to promote PRRSV replication as a negative regulator of type I IFN production [25,26,27]. However, miR-218, miR-23, and miR-26a could be used as a potential antiviral therapy for PRRSV due to their roles in promoting type I IFN responses [28,29,30]. Now, it is generally believed that miRNAs exert their biological functions in different ways to regulate type I IFN signaling. Therefore, it will be helpful to identify more functional miRNAs by exploring their roles in regulating type I IFN signaling.

In this study, we employed small RNA deep sequencing to identify differentially expressed miRNAs in PRRSV permissive cells (porcine alveolar macrophages, PAMs) and non-permissive cells (porcine peritoneal macrophages, PPMs), and found that miR-150 was highly expressed in PPMs. We then demonstrated that miR-150 expression was induced upon PRRSV infection via activating the PKC/JNK/c-Jun signaling pathway. Subsequently, we showed that miR-150 suppressed PRRSV replication through directly binding to its genome. In addition, we verified that miR-150 promoted type I IFN-triggered ISGs expression by suppressing SOCS1 expression. Therefore, we conclude that PRRSV infection upregulates miR-150 expression, which in turn inhibits PRRSV infection by targeting its genome and modulating type I IFN signaling.

## 2. Materials and Methods

### 2.1. Cells and Viruses

PAMs were obtained from lung lavage of 6–8-week-old specific pathogen-free (SPF) piglets (the Large White breed). PPMs were obtained from the peritoneal lavage of SPF pigs. Peripheral blood monocytes (PBMC) were isolated from SPF pigs by Ficoll-Paque (Sigma, Saint Louis, MO, USA) density gradient centrifugation according to the manufacturer’s instructions. PAMs, PPMs, and PBMC were cultured in RPMI 1640 (Gibco, Grand Island, NE,) with 10% heat-inactivated FBS (Gibco, Grand Island, NE, USA), 1% penicillin and streptomycin. Marc-145 cells were maintained in Dulbecco modified Eagle medium DMEM (Gibco, Grand Island, NE, USA) supplemented with 10% heat-inactivated FBS (Gibco, Grand Island, NE, USA), 1% penicillin and streptomycin. L929 cell culture supernatant was harvested as previously described [31]. All cells were maintained at 37 °C in an incubator with 5% CO_2_.

The highly pathogenic PRRSV (HP-PRRSV, PRRSV-2) (GenBank accession, JX317648) was propagated and titrated in PAMs and Marc-145 cells. The porcine epidemic diarrhea virus (PEDV) strain was propagated and titrated in Marc-145 cells. The viral supernatants from cell cultures were collected at different time points after virus inoculation, and the determination of viral 50% tissue culture infective doses (TCID_50_) was performed using the Reed–Muench method [32].

### 2.2. Small RNA Deep Sequencing

Total cellular RNA from PAMs and PPMs was isolated using the TRIzol reagent (Invitrogen, Carlsbad, CA, USA) to analyze the differential expression of miRNAs. Deep sequencing was performed using an Illumina Genome Analyzer at LC Sciences (Houston, TX, USA). The expression of miRNA was normalized and analyzed by calculating fold-change and *p*-value. miRNA was labeled as differentially expressed, when fold change was ≥2 and *p* ≤ 0.01.

### 2.3. Indirect Immunofluorescence Antibody (IFA) Assay

PAMs were transfected with miRNA mimics or negative controls (NC) for 12 h at the indicated concentration, followed by infection with PRRSV at an MOI of 0.01 for 36 h. The cells were fixed in cold methanol–acetone (1:1) for 10 min at 4 °C, washed with phosphate-buffered saline (PBS) three times, and then blocked with 10% goat serum in PBS for 30 min. Next, cells were incubated with anti-PRRSV N monoclonal antibody (prepared in our lab) at 37 °C for 1 h, washed with PBS three times, and followed by incubation with FITC-conjugated goat anti-mouse IgG antibodies at 37 °C for 1 h. After being washed with PBS three times, cells were examined with a fluorescence microscope.

### 2.4. Transfection of miRNA Mimic and Viral Infections

All the miRNA or NC mimics were transfected into PAMs at a concentration of 60 nM (except for the dose-dependent experiments) using HiPerfect transfection reagents (Qiagen, Dusseldorf, Germany). After transfection for 12 h, cells were infected with HP-PRRSV at an MOI of 0.01 unless otherwise stated. The cells and supernatants were then collected for quantitative real-time reverse transcription-PCR (qRT-PCR) and virus titer determination separately. miRNA and siRNA sequences are listed in Appendix A.

### 2.5. Inhibition of Signaling Transduction Pathways

Cells were pretreated with a PKC inhibitor (GF109203X, GF, 5 μM), JNK inhibitor (SP600125, SP, 10 μM), TAK-1 inhibitor (5Z-7-oxozeaenol, 5Z-7, 5 μM), NF-kB inhibitor (BAY11-7082, BAY, 1 μM), p38MAPK inhibitor (SB203580, SB, 10 μM), PI3K inhibitor (LY294002, LY, 5 μM), histone deacetylase inhibitor (TSA, 1 μM), DNA methyltransferase inhibitor (5-AZA-CdR, 5-AZA, 1 μM), c-Jun inhibitor (SR11302, SR, different doses), or DMSO control for 1 h, and then infected with or without HP-PRRSV at an MOI of 1 in the presence of inhibitors. The inhibitors were purchased from MCE (Monmouth Junction, NJ, USA). At 36 h post-infection, cells were harvested for miR-150 analysis by qRT-PCR or protein detection using Western blot.

### 2.6. RNA Extraction and Real-Time PCR

Total RNA was extracted from treated cells with TRIzon reagent (CW Biotech, Beijing, China) according to the manufacturer’s instructions. cDNA was prepared from 100 ng of RNA using the HiFiScript cDNA Synthesis Kit (CW Biotech, Beijing, China). For miRNA and primary-miRNA reverse transcription, cDNA was prepared using a primer set from Genepharma (Shanghai, China). qRT-PCR analysis was performed by using a ViiA 7 real-time PCR system (Applied Biosystems, Carlsbad, CA, USA) and SYBR green real-time PCR Master Mix (CW Biotech, Beijing, China). Gene qRT-PCR primers are listed in Appendix A. All qRT-PCR experiments were completed in triplicate.

### 2.7. Western Blot

PAMs were lysed with RIPA (CW Biotech, Beijing, China) supplemented with a proteinase and phosphatase inhibitor cocktail (Sigma, Saint Louis, MO, USA). Protein concentrations of the extracts were measured using a bicinchoninic acid (BCA) assay (Beyotime, Beijing, China). Proteins were separated on SDS-PAGE gels and transferred onto polyvinylidene difluoride (PVDF) membranes (Millipore, Billerica, MA, USA). Membranes were blocked with 5% skim milk in PBS with 0.1% Tween-20 for 1 h at room temperature, and then incubated with anti-PRRSV N, anti-phospho-JNK, anti-phospho-c-Jun, anti-JNK, anti-c-Jun, or anti-SOCS1 Abs at 4 °C overnight. These antibodies were purchased from Cell Signaling Technology (Boston, MA, USA). The membranes were then incubated with HRP-conjugated anti-rabbit/mouse secondary Ab (Solarbio, Beijing, China) for 1 h at room temperature. β-actin was used as a loading control. The immunolabeled proteins were visualized using ECL reagent (CW Biotech, Beijing, China) following the manufacturer’s instructions.

### 2.8. RNA Immunoprecipitation Assay

PAMs were transfected with NC or miR-150 mimics using HiPerfect reagents (Qiagen, Dusseldorf, Germany), subsequently infected with the HP-PRRSV at an MOI of 0.5 for 36 h. Cells were harvested and lysed using cell lysis buffer (1% NP-40, 150 mM NaCl, 10 mM Tris-HCl, pH 7.8 with 1 mM EDTA) containing an RNase inhibitor (1:100). The supernatants were incubated with an anti-Ago2 monoclonal antibody (Abnova, Taiwan, China) or isotype control IgG at 4 °C overnight and then mixed with protein A-agarose (Sigma, Saint Louis, MO, USA) for 2 h. The RNA from the immunoprecipitation product was isolated. The expression levels were quantified by qRT-PCR. Gene qRT-PCR primers are listed in Appendix A.

### 2.9. Plasmid Construction and Dual Luciferase Reporter Assays

The 2 kb regions upstream of the pre-miR-150 precursor gene were determined as the promoter region using the Ensembl Genome Browser (http://asia.ensembl.org/index.html, accessed on 10 March 2021). A 2-kb-length miR-150 gene promoter was amplified from PAM DNA and cloned into the luciferase reporter vector pGL3-Basic. The truncated mutants of the miR-150 promoter were constructed using the primers listed in Appendix A and inserted into the pGL3-basic vector. The porcine SOCS1 3′UTR and HP-PRRSV 3′UTR were amplified and inserted into the pGL3-control vector. c-Jun binding element deletion mutants and the target site mutants were generated using the Q5 site-directed mutagenesis kit (NEB). Gene PCR primers are listed in Appendix A.

Marc-145 cells were cotransfected with pGL3 luciferase reporter vector, pRL-TK, or miRNAs for 36 h Cell extracts were prepared and analyzed for firefly and Renilla luciferase activities using a dual-luciferase reporter assay kit (Promega, Madison, WI, USA) according to the manufacturer’s instructions.

### 2.10. Chromatin Immunoprecipitation (ChIP) Assay

ChIP assays were performed by using a SimpleChIP^®^ Enzymatic Chromatin IP Kit (Cell Signaling Technologies, Boston, MA, USA) according to the manufacturer’s instructions. PAMs were mock infected or infected with HP-PRRSV strain (MOI = 1) for 36 h. For optimal ChIP results, at least 4 × 10^6^ cells are required for each immunoprecipitation. Briefly, PAMs were treated with formaldehyde for protein–DNA crosslinking and then collected in the lysis buffer. DNA was sheared to an average size of 150–900 bp using nuclease and sonication. Sheared DNA (5–10 μg) was incubated with 5 μg of the anti-p-c-Jun antibody (Cell Signaling Technologies) or normal rabbit IgG at 4 °C overnight, followed by incubation with 30 μL protein G Magnetic Beads. The enriched DNA was processed by proteinase K digestion and purified with spin columns. Immunoprecipitated DNA was analyzed by PCR. PCR primers are listed in Appendix A.

### 2.11. Statistical Analysis

Statistical analysis was performed using GraphPad Prism software, and differences were analyzed using Student’s *t*-test. Significance is denoted in the figures as follows: *, *p* < 0.05; **, *p* < 0.01; ***, *p* < 0.001; and ns, not significant.

## 3. Results

### 3.1. miR-150 Is Induced by PRRSV Infection

PRRSV has a restricted cell tropism and infects cells from the monocyte/macrophage lineage in their natural hosts. We performed small RNA deep sequencing to find DEmiRNAs in PAMs and PPMs and identified that miR-150 was constitutively expressed higher in PPMs than that in PAMs. To confirm this observation, we performed qRT-PCR and found that the expression level of miR-150 was indeed much higher in PPMs (~5-fold) than that in PAMs (Figure 1A). Since PBMC are hardly permissive to PRRSV infection, we next examined whether miR-150 was highly expressed in PBMC. As shown in Figure 1A, our results showed that miR-150 was ~10-fold higher in PBMC than that in PAMs. As PBMC differentiate into macrophages, they are becoming permissive to PRRSV infection. Thus, we cultured PBMC in culture media supplemented with culture supernatant of L929 cells, and then analyzed miR-150 expression at different culture time points. Our results showed that miR-150 expression was gradually decreased during monocyte–macrophage differentiation (Figure 1B). Since miRNAs could be altered in virus-infected cells, we evaluated miR-150 expression in PRRSV-infected PAMs at different times post infection using qRT-PCR. Our results revealed that miR-150 was significantly increased by ~2- and ~2.7-fold at 36 and 48 h post infection (hpi), respectively (Figure 1C). Additionally, its primary transcript (primary miR-150) was also enhanced by ~2.4- and ~3.2-fold at 36 and 48 hpi, respectively (Figure 1D). As shown in Figure 1E,F, both miR-150 and primary miR-150 were significantly induced by PRRSV in a dose-dependent manner. Taken together, these results uncover that PRRSV infection induces miR-150 expression.

### 3.2. PRRSV Induces miR-150 Expression by Activating the PKC/JNK/c-Jun Pathway

Regulating miRNA expression is one of the evolved strategies for hosts to defend against virus infection. To explore the underlying molecular mechanism about how PRRSV regulates miR-150 expression, we pretreated PAMs with different canonical signaling pathway inhibitors and epigenetic modification inhibitors of methylation and deacetylation. As shown in Figure 2A, PRRSV-induced miR-150 expression was suppressed to 45% and 50% by the PKC inhibitor (GF109203X) and JNK inhibitor (SP600125) compared to DMSO, respectively. The PKC family converts environmental changes into cellular actions through multiple signal transduction networks [33,34]. The mitogen-activated protein kinase (MAPK) composed of JNKs, the extracellular regulated kinases (ERKs) and p38 MAPKs, are essential in virus infection. PKC can phosphorylate JNK and enhance JNK activation [35,36], and c-Jun is downstream of JNK [37]. Thus, to better understand the signaling pathways involved in PRRSV-induced miR-150 expression, we pretreated PAMs with PKC, JNK, or c-Jun inhibitors. Our data showed that miR-150 and primary miR-150 in PRRSV-infected PAMs were suppressed by these inhibitors in a dose-dependent manner, implying that miR-150 expression might be regulated via the PKC/JNK/c-Jun pathway (Figure 2B–D). To further validate the pathway regulating miR-150 expression, PAMs were pretreated with the PKC inhibitor. Western blot analysis showed that the PKC inhibitor repressed JNK and c-Jun phosphorylation (Figure 2E). These data suggest that PRRSV upregulated the expression of miR-150 via activating the PKC/JNK/c-Jun signaling pathway.

### 3.3. c-Jun Is Crucial for PRRSV to Activate the miR-150 Promoter

To further investigate the regulation of miR-150 by PRRSV, we constructed a miR-150 promoter vector and a series of truncated mutants with deletions starting from the 5′ end of the promoter. Our results validated that the luciferase activity of the miR-150 promoter was enhanced by PRRSV infection in a dose-dependent manner (Figure 3A). Additionally, the luciferase activities of −1524/+25-Luc, −1014/+25-Luc, and −500/+25-Luc mutants were enhanced by PRRSV infection, while −43/+25-Luc did not respond to PRRSV infection, suggesting that there might be important cis-activating elements existing in the region from −500 to −43 bp (Figure 3B). Then, we analyzed the region from −500 to −43 bp using bioinformatics tools PROMO 2.0 (http://www.cbs.dtu.dk/services/Promoter/, accessed on 22 April 2021) and found that there were two putative c-Jun binding sites (−438 to −433 bp and −135 to −130 bp). To confirm the role of the transcription factor c-Jun in the upregulation of miR-150, we investigated the luciferase activities of mutants with the deletion of either c-Jun binding site upon PRRSV infection (Figure 3C). The results showed that both deletion mutants lost their ability to respond to PRRSV infection, indicating that the transcription factor c-Jun is essential for PRRSV to activate miR-150 promoter (Figure 3D). Knockdown of c-Jun significantly impaired PRRSV-induced miR-150 expression in a dose-dependent manner (Figure 3E,F). Furthermore, to analyze the DNA–protein interaction between miR-150 promoter and c-Jun, we performed a chromatin immunoprecipitation assay (ChIP). As shown in Figure 3G, c-Jun could directly bind miR-150 promoter in the sites of −438 to −433 bp and −135 to −130 bp. Taken together, these results indicate that c-Jun is a critical transcriptional factor for PRRSV to activate the miR-150 promoter and upregulate miR-150 expression.

### 3.4. miR-150 Inhibits PRRSV Replication

To investigate whether miR-150 regulates PRRSV replication, we transfected miR-150 mimics into PAMs before PRRSV infection. The immunofluorescence assay showed that overexpression of miR-150 inhibited PRRSV replication in a dose-dependent manner (Figure 4A). Moreover, qRT-PCR results uncovered that PRRSV ORF7 mRNA level was also impaired by miR-150 (Figure 4B). However, when the endogenous miR-150 was downregulated by its inhibitor, PRRSV ORF7 mRNA level displayed a significant increase (Figure 4C). Next, we investigated the impact of miR-150 on PRRSV replication at different times post infection. As shown in Figure 4D, ORF7 mRNA level was suppressed by miR-150 at different times post PRRSV infection and the inhibitory effect peaked at 48 hpi. In addition, Western blot analysis showed that PRRSV N protein level was also significantly reduced by miR-150 (Figure 4E). To investigate the effect of miR-150 on the dynamics of PRRSV growth, we measured the virus titer using the TCID_50_ assay. Viral growth was markedly suppressed about 30-fold by miR-150 at 60 hpi (Figure 4F). These data indicate that miR-150 suppresses PRRSV replication.

### 3.5. miR-150 Directly Targets the PRRSV Genome

To investigate the molecular mechanism behind the regulation of PRRSV infection by miR-150, we first examined whether the HP-PRRSV HV strain had the miR-150 target site in its genome using RegRNA and ViTa. Our analysis revealed that there was a putative miR-150 target site in PRRSV 3′UTR (15,205–15,212 bp) (Figure 5A). For conservation analysis, we aligned target sequences in 284 genotype 2 strains and 22 genotype 1 strains. The results showed that target region was moderately conserved in type 2 PRRSV (78%) and highly conserved in the 55 HP-PRRSV isolates (91%). However, miR-150 does not target PRRSV type 1 strains. To verify that miR-150 targets PRRSV 3′UTR, we performed a luciferase reporter assay. The luciferase activity of WT 3′UTR was significantly suppressed by miR-150 in a dose-dependent manner (Figure 5B). However, miR-150 did not inhibit the luciferase activity of Mut 3′UTR with the mutated seed sequence (Figure 5C). These data suggest that miR-150 targets PRRSV. To further elucidate the viral RNA and miR-150 interaction, we applied an RNA-induced silencing complex (RISC) immunoprecipitation assay. Our results verified that viral RNA was enriched by approximately 2-fold compared to NC, indicating that PRRSV RNA physically binds to the RISC complex (Figure 5D). miR-150 was also significantly enriched about 150-fold (Figure 5E). Overall, these results suggest that miR-150 suppresses PRRSV replication by directly targeting its 3′UTR.

### 3.6. miR-150 Facilitates Type I IFN Responses by Targeting SOCS1

To assess whether miR-150 regulates antiviral response by targeting other vital factors, we predicted the potential targeting genes using RNAhybrid, TargetScan, and RNAcentral bioinformatics programs. Our analysis uncovered that SOCS1 3′UTR had an atypical complimentary site with miR-150 and the target site was conserved in mammals (Figure 6A). Previous reports have shown that SOCS1, a negative factor to type I IFN responses, is induced by PRRSV to facilitate immune evasion [38]. To provide direct evidence that miR-150 could target SOCS1, we performed a luciferase reporter assay. Our results showed that miR-150 repressed the luciferase activity of SOCS1 3′UTR to 60% compared to NC, whereas the miR-150 inhibitor resulted in a ~2-fold increase (Figure 6B). When the target site was mutated, the inhibitory effect of miR-150 was abolished, indicating that miR-150 targets SOCS1 3′UTR (Figure 6C). Next, we applied an RISC-IP assay to explore the interaction between miR-150 and SOCS1. The qRT-PCR data revealed that miR-150 enhanced SOCS1 mRNA bound to RISC by ~2.4-fold, suggesting that miR-150 physically binds to SOCS1 3′UTR (Figure 6D). Taken together, we conclude that SOCS1 is a direct target of miR-150.

To assess the effect of miR-150 on antiviral responses by targeting SOCS1, we evaluated whether miR-150 promoted type I IFN responses. As shown in Figure 6E,F, SOCS1 mRNA was significantly downregulated by miR-150 mimics upon PRRSV infection, but promoted by the miR-150 inhibitor. Downstream ISGs such as Mx1 and ISG56 mRNA were increased by miR-150. However, the miR-150 inhibitor led to an opposite result that Mx1 and ISG56 mRNAs were decreased by 20% and 40%, respectively. When PAMs were stimulated with poly(I:C), miR-150 also facilitated ISGs expression via targeting SOCS1 (Figure 6G). However, there was a significant increase in SOCS1 mRNA when the miR-150 inhibitor was applied, and downstream ISGs expression levels were repressed by the miR-150 inhibitor upon poly(I:C) stimulation (Figure 6H). Next, we investigated whether miR-150 suppressed SOCS1 protein level upon PRRSV infection. SOCS1 and PRRSV N protein levels were downregulated by miR-150, which was similar to the results with knockdown of SOCS1 by siRNA-SOCS1 (Figure 6I). These data indicate that miR-150 facilitates type I IFN responses by suppressing SOCS1 expression; therefore, resulting in the inhibition of PRRSV replication.

### 3.7. miR-150 Has Inhibition of Porcine RNA Virus

Since miR-150 could promote type I IFN responses by targeting SOCS1, we further investigated whether miR-150 has a broad-spectrum antiviral property for other viruses. Given that PEDV belongs to the same order *Nidovirales* with PRRSV, we then tested whether miR-150 inhibited its infection. The qRT-PCR results showed that PEDV replication was inhibited to 50% by miR-150 (Figure 7A). Together, miR-150 and primary miR-150 were also induced after PEDV infection at 36 hpi (Figure 7B,C). Thus, miR-150 can be induced upon infection of PEDV and inhibit its replication.

## 4. Discussion

Hosts exert multiple mechanisms to regulate PRRSV infection. Here, we found that miR-150 existed at a relatively low level in PRRSV permissive cells (PAMs) but was induced upon PRRSV infection. Subsequently, we demonstrated that PRRSV upregulated miR-150 expression via the activation of the PKC/JNK/c-Jun signaling pathway. Finally, we showed that miR-150 inhibited PRRSV replication by targeting the viral genome and enhancing type I IFN signaling by targeting SOCS1.

PRRSV shows differential infectivity to different host cells and tissues, and tends to infect cells from monocyte–macrophage lineages, such as PAMs [39]. Different cells and tissues exhibit a distinct miRNA expression profile [40]. Additionally, these different expression patterns might be partly responsible for the tropism of PRRSV. It has been shown that miR-181 expression is negatively correlated with PRRSV replication capacity in different cells, suggesting that miR-181 might play a role in determining PRRSV tropism [41]. Another mode of viral tropism regulation by cellular miRNA occurs in HCV, and its replication is promoted by liver-specific miRNA miR-122 [35,42]. We explored DEmiRNAs between PAMs and PPMs using small RNA deep sequencing and identified miR-150 as a candidate, which was shown to be gradually decreased during the differentiation of PBMC. Interestingly, it has been reported that cultured PBMC becomes much more permissive to PRRSV infection [31]. Combined with the results that miR-150 can be upregulated by PRRSV, we assume that miR-150 may affect PRRSV replication.

The research on how miRNAs exert their multiple cellular functions has gained much attention. However, the question of how miRNAs are regulated needs to be adequately addressed [43,44,45]. We found that PRRSV activates the PKC/JNK/c-Jun signaling pathway to upregulate miR-150 expression. Additionally, the transcription factor c-Jun is essential to regulate miR-150 expression upon PRRSV infection. As a serine/threonine kinase, PKC regulates multiple cellular processes, including regulation of viral replication through facilitating phosphorylation of proteins [46,47]. In response to HCV infection, the viral protein NS4B activates PKC, and then stimulates ERK/JNK cascades, resulting in the regulation of cell transformation and apoptosis [35]. PKCδ is demonstrated to play an important role in facilitating PRRSV replication at early steps of viral replication [48]. While different PKC isoforms are involved in different signaling pathways [49], the PKC different isoforms inhibitor GF109203X suppressed PKC/JNK/c-Jun cascades to inhibit miR-150 expression in our study.

To date, the interplay between miRNA and PRRSV is well studied. miR-181 and miR-124a can target the PRRSV receptor CD163; thus, suppressing PRRSV replication [50,51]. Another PRRSV entry receptor CD151 can be regulated by miR-506 [52]. The mechanism of miRNA regulating PRRSV entry provides new insights into understanding PRRSV cell tropism. In addition, the PRRSV genome can be targeted by many miRNAs, including miR-23 and miR-331-3p, which can reveal another mechanism how miRNAs regulate PRRSV replication [29,53]. miRNA can also regulate host antiviral responses. Besides affecting type I IFN signaling, miR-22 promotes PRRSV replication through targeting host factor HO-1 [54]. Additionally, miR-376b-3p, as a pro-viral miRNA, was involved in the lysosomal pathway by targeting TRIM22 [55]. Based on numerous studies, miRNAs have the potential use to control PRRSV.

The significance and underlying mechanism for miR-150 to inhibit PRRSV replication was elucidated in our study. Directly targeting PRRSV 3′UTR by miR-150 affected viral RNA replication. miR-150 was also shown to be involved in promoting type I IFN signaling; therefore, playing a vital role in antiviral immunity. Type I IFNs induce numerous antiviral genes through activating the JAK/STAT pathway [56]. Thus, when exploring the function of miRNAs on virus replication, one of the most obvious directions is whether miRNAs modulate innate immune responses. SOCS1 functions as a negative feedback regulation of cytokine-induced signaling pathway by directly binding to JAKs and inhibiting their catalytic activity [57,58]. Previous reports have shown that suppression of SOCS1 is correlated with increased miR-150 expression level in DHF (Dengue hemorrhagic fever) patients [59]. miR-150 is also a potential therapy for renal tubulointerstitial fibrosis, due to its effect on suppressing the JAK/STAT pathway by downregulating SOCS1 [60,61]. In addition, Luo et al. have shown that PRRSV can evade host innate immune responses through inducing SOCS1 to inhibit the expression of IFN and ISGs [38]. For maintaining homeostasis, hosts may exploit various strategies, including upregulating miRNAs. miR-150 was induced more obviously at the late stage of PRRSV infection. The upregulation of SOCS1 and miR-150 by PRRSV might reflect the counter reaction between PRRSV and host. However, the underlying mechanism is unclear and needs further study. Modulation of SOCS1 seems to be a way for the virus and host to counter with each other. Here, we demonstrated that miR-150 was induced during PRRSV infection, and in turn miR-150 enhanced type I interferon function by downregulating SOCS1; thus, inhibiting virus replication.

Since miR-150 has effects on type I IFN signaling, we assume that miR-150 might have broad-spectrum antiviral activity. Indeed, we showed that miR-150 could inhibit PEDV replication, which belongs to the same order *Nidovirales* with PRRSV. Previous studies have also shown that miR-150 is upregulated by HIV infection and is used as a novel therapeutic target against HIV [62,63,64,65]. Moreover, miR-150, which is highly expressed in in patients with severe A/H1N1 disease, is also shown to inhibit H1N1 replication through targeting the PB2 gene [66,67]. miR-150 also targets SARS-CoV-2 nsp10 to suppress its infection [68]. As a virus inducible regulator, miR-150 seems to have a broad-spectrum antiviral ability [69].

In summary, we found that miR-150 was upregulated upon PRRSV infection via activating the PKC/JNK/c-Jun pathway. Then, we verified that miR-150 acted as a negative regulator of PRRSV replication by directly targeting the PRRSV genome and regulating I-IFN signaling by targeting SOCS1. Hence, miR-150 may be a potential therapeutic agent against PRRSV infections (Figure 8).

## Figures and Tables

**Figure 1 viruses-14-01485-f001:**
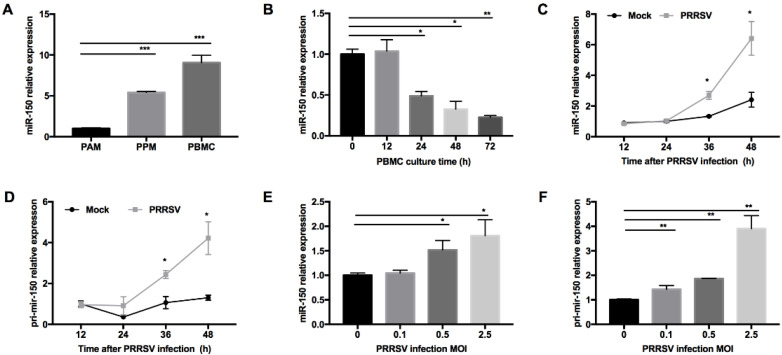
miR-150 expression is induced by PRRSV infection. (**A**) miR-150 expression level in PAMs, PPMs, and PBMC was analyzed by qRT-PCR. (**B**) PBMC was cultured in the presence of L929 cell culture supernatants, and total RNA was extracted at 0, 12, 24, 36, 48, and 72 h for miR-150 analysis by qRT-PCR. (**C**,**D**) PAMs were inoculated with medium alone or HV-PRRSV (HV isolate) at an MOI of 0.5. Total RNA was extracted at 12, 24, 36, and 48 hpi. qRT-PCR was used to analyze miR-150 and primary miR-150 expression. (**E**,**F**) PAMs were either mock or infected with HP-PRRSV (HV isolate) at an MOI of 0.1, 0.5, or 2.5 for 36 h, and total RNA was extracted for detecting miR-150 and primary miR-150 expression by qRT-PCR. The data are representative of three independent experiments (means ± SEM). *p* values were analyzed using a *t*-test. *, *p* < 0.05; **, *p* < 0.01; ***, *p* < 0.001.

**Figure 2 viruses-14-01485-f002:**
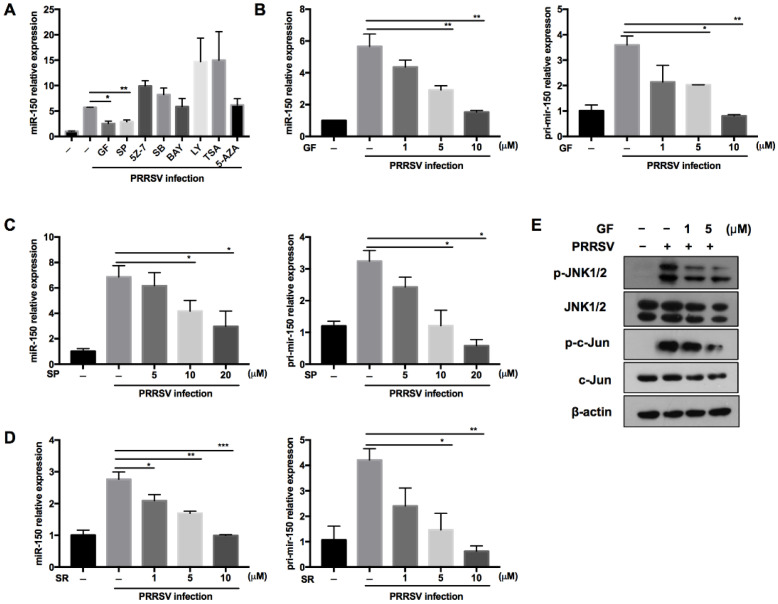
PRRSV infection induces miR-150 expression via the PKC/JNK/c-Jun pathway. (**A**) PAMs were pretreated for 1 h with inhibitors of the PKC inhibitor (GF), JNK inhibitor (SP), TAK-1 inhibitor (5Z-7), NF-kB inhibitor (BAY), p38MAPK inhibitor (SB), PI3K inhibitor (LY), histone deacetylase inhibitor (TSA), DNA methyltransferase inhibitor (5-AZA), or DMSO control, and then inoculated with HP-PRRSV (HV isolate) (MOI = 1). At 36 hpi, miR-150 expression was analyzed by qRT-PCR. (**B**–**D**) PAMs were pretreated for 1 h with the PKC inhibitor (GF), JNK inhibitor (SP), and c-Jun inhibitor (SR11302, SR) at different doses, and then infected with HP-PRRSV (HV isolate) (MOI = 1). At 36 hpi, total RNAs were extracted for analyzing miR-150 and pri-miR-150 expression by qRT-PCR. (**E**) PAMs were pretreated with the PKC inhibitor (GF) for 1 h, followed by infection with HV-PRRSV (HV isolate) (MOI = 1), and cells were harvested at 24 hpi. Western blot was used to examine the expression levels of p-JNK, total-JNK, p-c-Jun, total-c-Jun, and β-actin. The data are representative of three independent experiments (means ± SEM). *p* values were analyzed using a *t*-test. *, *p* < 0.05; **, *p* < 0.01; ***, *p* < 0.001.

**Figure 3 viruses-14-01485-f003:**
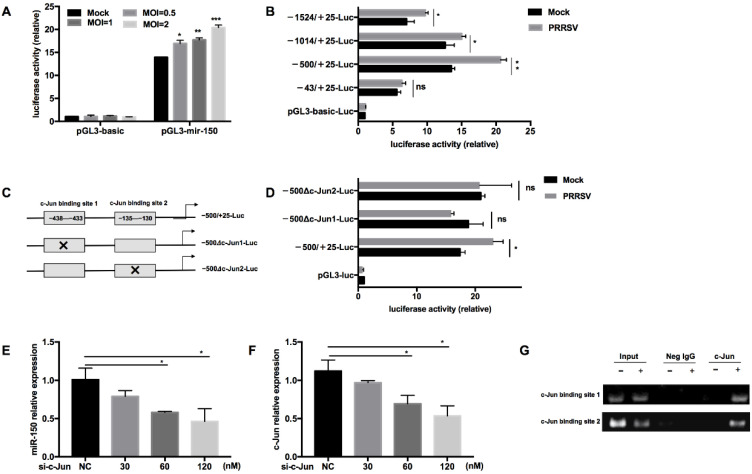
c-Jun is crucial for PRRSV to activate the miR-150 promoter. (**A**,**B**) After co-transfection of pGL3-miR-150 or truncated pGL3-miR-150 along with pRL-TK into Marc-145 cells for 12 h, cells were infected with HV-PRRSV (HV isolate) at an MOI of 1 or indicated MOIs. At 36 hpi, cells were then harvested to assess the luciferase activity. (**C**) Schematic representation of the miR-150 promoter c-Jun-deleted mutant vectors. (**D**) Marc-145 cells were co-transfected with WT or deletion mutant vectors along with pRL-TK for 12 h and then infected with HV-PRRSV (HV isolate) (MOI = 1). At 36 hpi, cells were then harvested to assess the luciferase activity. (**E**,**F**) PAMs were transfected with siRNA targeting c-Jun, and then the cells were inoculated with or without HV-PRRSV (HV isolate) (MOI = 0.1). At 36 hpi, total RNA was extracted for c-Jun and miR-150 expression analysis. (**G**) PAMs were infected mock or HV-PRRSV (HV isolate) at an MOI of 1 for 36 h. A ChIP assay was then performed with anti-p-c-Jun antibody, or negative control normal rabbit IgG. The precipitated miR-150 promoter was amplified by PCR. The data are representative of three independent experiments (means ± SEM). *p* values were analyzed using a *t*-test. *, *p* < 0.05; **, *p* < 0.01; ***, *p* < 0.001.

**Figure 4 viruses-14-01485-f004:**
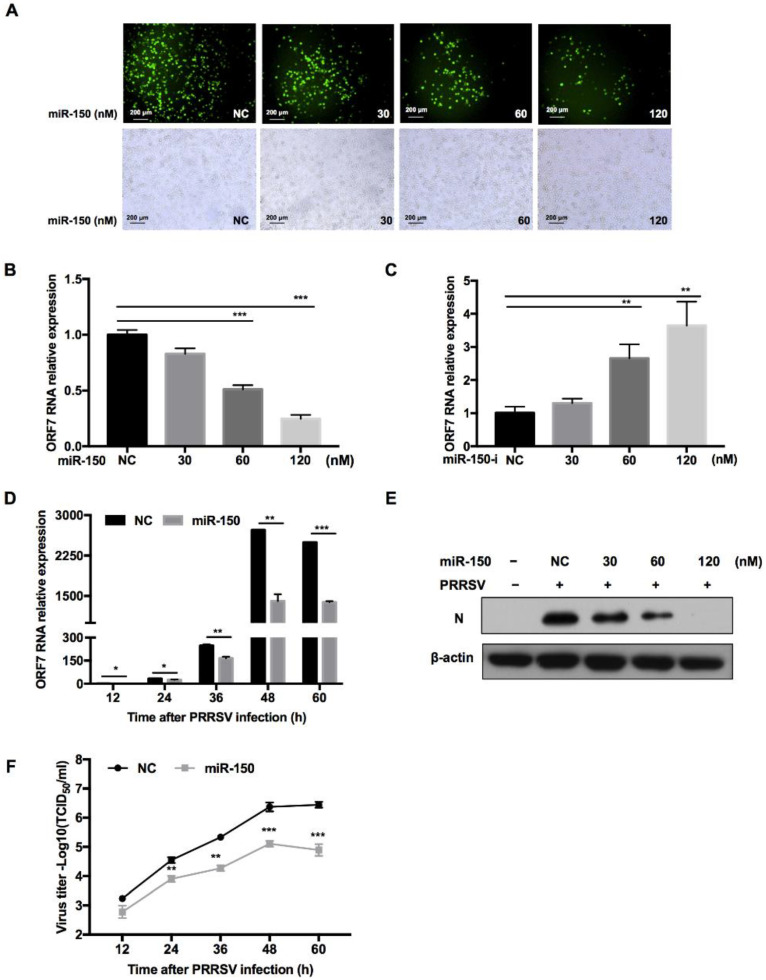
miR-150 inhibits PRRSV replication. PAMs were transfected with miR-150 mimics or miR-150 inhibitors (miR-150-i) at the indicated concentration for 12 h, followed by infection with HV-PRRSV (HV isolate) for 36 h at an MOI of 0.01. (**A**) Cells were then fixed for immunofluorescent staining of PRRSV N protein. (**B**,**C**) qRT-PCR was used to analyze ORF7 expression. (**E**) Cells were harvested for PRRSV protein N detection using Western blot analysis. (**D**,**F**) PAMs were transfected with miR-150 mimics or negative control (NC) at a final concentration of 60 nM, followed by inoculation with medium alone, or HP-PRRSV (HV isolate) at an MOI of 0.01. Total RNA was extracted at 12, 24, 36, 48, and 60 h, respectively. qRT-PCR was used to analyze ORF7 expression (**D**). Culture supernatants were collected to analyze virus yields (**F**). The data are representative of three independent experiments (means ± SEM). *p* values were analyzed using a *t*-test. *, *p* < 0.05; **, *p* < 0.01; ***, *p* < 0.001.

**Figure 5 viruses-14-01485-f005:**
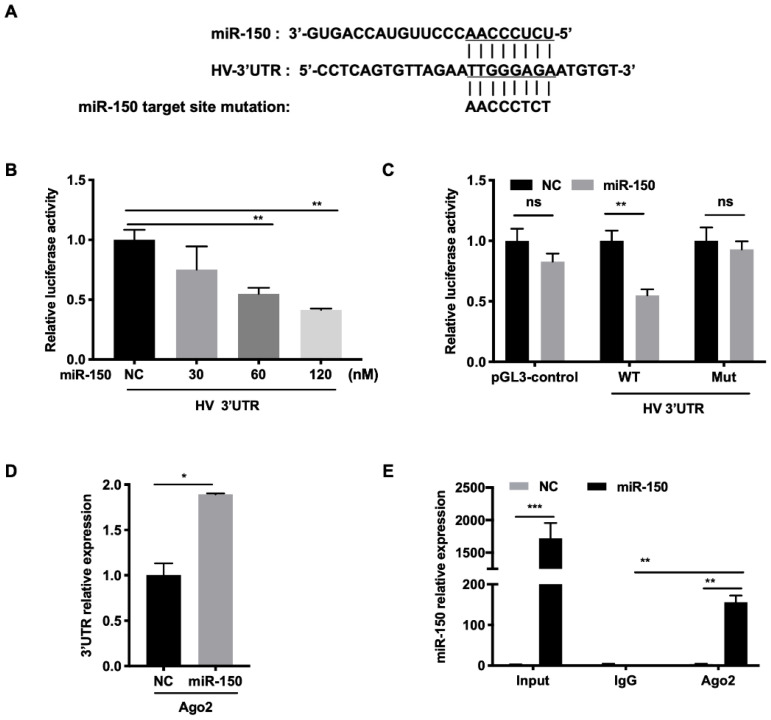
miR-150 directly targets the PRRSV genome. (**A**) Schematic diagram of the predicted target sites of miR-150 in HV-PRRSV (HV isolate) 3′ UTR. The predicted target sites of miR-150 were mutated as indicated. (**B**,**C**) Marc-145 cells were co-transfected with HV 3′ UTR or HV 3′ UTR mutant luciferase reporter vector, pRL-TK, and miR-150 mimics or miR-150 inhibitor (miR-150-i) for 36 h. Cells were then harvested for luciferase assay. (**D**,**E**) PAMs were transfected with miR-150 mimics or NC at a final concentration of 60 nM for 12 h, followed by infection with HV-PRRSV (HV isolate) for 36 h at an MOI of 0.05. qRT-PCR was used to analyze HV-PRRSV (HV isolate) 3′ UTR among RNA extracted from RISC immunoprecipitates of the cells and miR-150 among RNA extracted from total samples, RISC immunoprecipitates, or IgG (isotype control) immunoprecipitates of lysates. The data are representative of three independent experiments (means ± SEM). *p* values were analyzed using a *t*-test. *, *p* < 0.05; **, *p* < 0.01, ***, *p* < 0.001.

**Figure 6 viruses-14-01485-f006:**
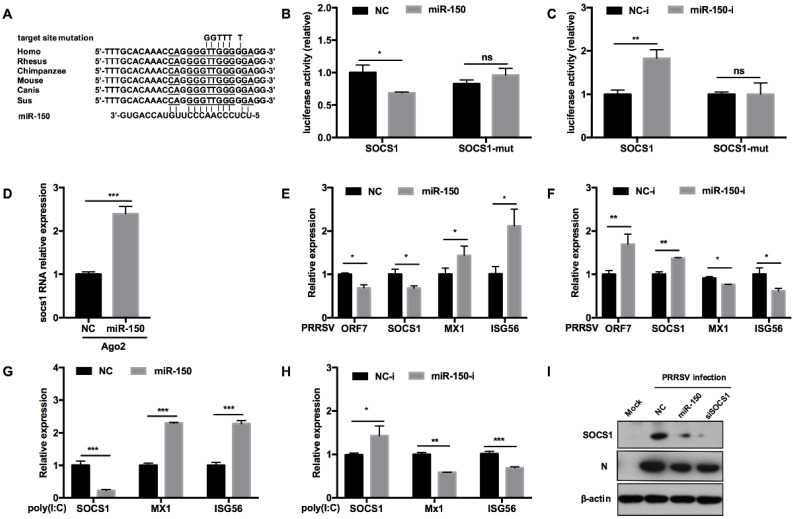
miR-150 promotes type I IFN responses by targeting SOCS1. (**A**) Schematic diagram of the predicted target sites of miR-150 in SOCS1 3′ UTR and target sites of six representative mammals. The predicted target sites of miR-150 were mutated as indicated. (**B**,**C**) Marc-145 cells were co-transfected with SOCS1 3′ UTR or SOCS1 3′ UTR mutant luciferase reporter vector, pRL-TK, and miR-150 mimics or miR-150 inhibitor (miR-150-i) for 36 h and then harvested for luciferase assay. (**D**) qRT-PCR analysis of SOCS1 3′ UTR among RNAs extracted from RISC immunoprecipitates of the cells, which is indicated in Figure 5D. (**E**–**H**) PAMs were transfected with miR-150 mimics or miR-150 inhibitors (miR-150-i) at a final concentration of 60 nM for 24 h, followed by infection with HV-PRRSV (HV isolate) (MOI = 0.5) for 24 h or stimulating with poly(I:C) (10 μg/mL) for 24 h. Total RNA was extracted and qRT-PCR was used to analyze ORF7, SOCS1, Mx1, and ISG15 expression. (**I**) PAMs were transfected with miR-150 mimics, siRNA-SOCS1 (an equimolar mixture of two siRNAs), or NC at a final concentration of 60 nM for 24 h, followed by infection with HV-PRRSV (HV isolate) for 36 h at an MOI of 0.5. Cells were harvested for PRRSV protein N and SOCS1 protein detection using Western blot analysis. The data are representative of three independent experiments (means ± SEM). *p* values were analyzed using a *t*-test. *, *p* < 0.05; **, *p* < 0.01; ***, *p* < 0.001.

**Figure 7 viruses-14-01485-f007:**
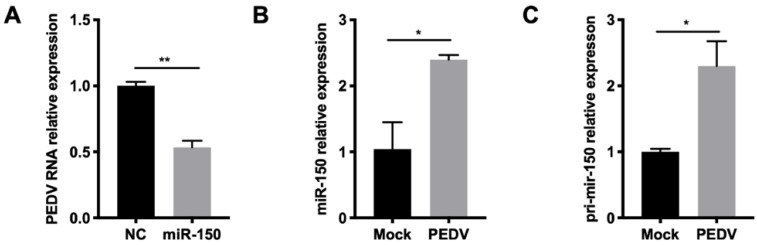
miR-150 has a broad-spectrum antiviral property for RNA virus. (**A**,**B**) Marc-145 cells were infected with PEDV at an MOI of 0.5 for 36 h. Total RNA was extracted, and qRT-PCR was used to analyze miR-150 and pri-miR-150 expression. (**C**) Marc-145 cells were transfected with miR-150 mimics or NC at a final concentration of 60 nM for 12 h, followed by infection with PEDV for 36 h at an MOI of 0.1. qRT-PCR was used to analyze PEDV RNA expression. The data are representative of three independent experiments (means ± SEM). *p* values were analyzed using a *t*-test. *, *p* < 0.05; **, *p* < 0.01.

**Figure 8 viruses-14-01485-f008:**
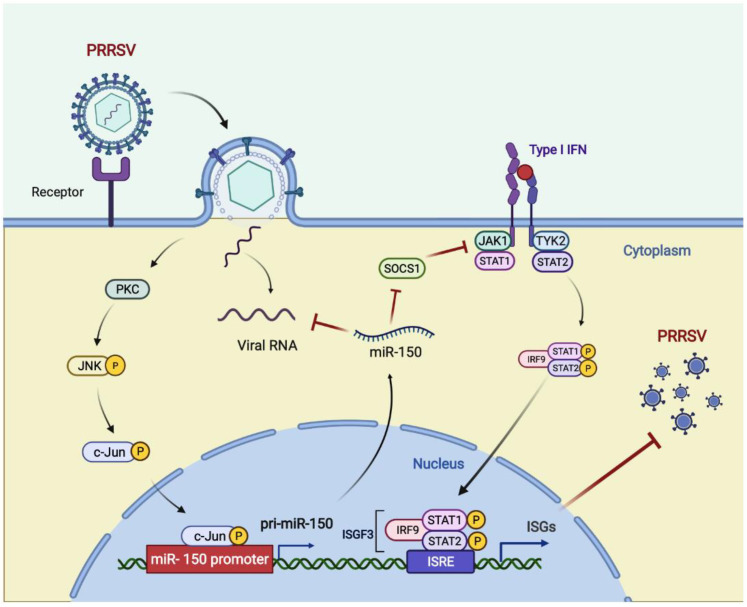
Model for the induced miR-150 to regulate PRRSV replication. Upon PRRSV infection, host cells upregulate c-Jun-mediated miR-150 expression as a host defensive strategy. miR-150 feedback suppresses PRRSV replication by targeting the viral genome. SOCS1-mediated suppression of ISGs is alleviated by miR-150; thus, suppressing PRRSV replication. Figure created with BioRender (https://biorender.com, accessed on 2 April 2022).

## Data Availability

The datasets generated for this study are available on request to the corresponding author.

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
