# Peer review of "Inducible miR-150 Inhibits Porcine Reproductive and Respiratory Syndrome Virus Replication by Targeting Viral Genome and Suppressor of Cytokine Signaling 1"

_viruses, 2022, doi:10.3390/v14071485_

Round 1

Reviewer 1 Report

By comparing the small RNA deep sequencing results of PRRSV tropic and non-tropic cells, the authors took miR-150 as the research object, and found that PRRSV induced the expression of miR-150 since PRRSV could activate the PKC/JNK/c-jun signaling pathway, and found that miR-150 inhibited PRRSV replication by targeting SOCS1 to activate the IFN-I signaling pathway and by targeting the PRRSV gene. This result is very interesting, and these results will provide a theoretical basis for PRRSV prevention and control. However, the manuscript must be improved before it can be accepted.

Major problem:

1.      Introduction:

1.1 The nomenclature of PRRSVs must be updated, to include both species, with an appropriate recent reference.

1.2 the description of type I interferon signal pathway is too brief and needs a brief description.

2.      Materials and methods:

2.1 The breed of pig should be indicated in the Materials and Methods

2.2 It should be specified whether the virus is PRRSV-1 or PRRSV-2.

2.3 “the determination of viral 50% tissue culture infective doses (TCID50) was performed by the Reed-Muench method” should be added appropriate recent references

3.      Results:

3.1   The authors claim that "differentially expressed miRNAs (DEmiRNAs) in PPM and PAM may play a role in regulating PRRSV infection"(on lines 171-172), however, the authors do not show the sufficient evidence. For example, the authors did not show the viral titers of PRRSV in these two cells; nor do they show the disproportionate result that If the miR-150 of PPM is artificially down-regulated, whether it is the same as that of PAM, PPM becomes a susceptible cell; the discussion section does not discuss the relevant data of PPM and PAM in PRRSV tropism. Therefore, all parts of the full text of the manuscript need to delete this statement or change it.

3.2    It has been reported that PRRSV can induce SOCS1, while the authors have found that PRRSV induces miR-150 and the latter targets SOCS1. The author needs to explain its mechanism in Discussion. If there is no clear mechanism, please specify that further research is needed in Discussion.

3.3    On lines212-222:The author needs to briefly describe the PKC/JNK/c-Jun signal pathway, at least let the reader understand the significance of its phosphorylation, and then naturally come to the conclusion that PRRSV may activate the signal pathway.

3.4    Figure 3G needs to adjust the contrast to make the stripes clearer.

3.5    The ordinate mark in figure 4 is wrong, that is, the log should be preceded by a minus sign, and the TCID is 50 less.

3.6   On line 370: “miR-150 has a broad-spectrum inhibition of porcine RNA virus” is not suitable in Results Just because it has done the inhibitory effect on PEDV. However, after enumerating many examples in lines 440-448 of the Discussion, this expression becomes reasonable. So, the authors need to revise it in Restuts.

4.      Discussion:

Considering the powerful regulatory function of miRNA and the large number of studies in PRRSV in recent years, it is recommended that the author list relevant literatures on classification, discuss the research progress of PRRSV and miRNA in detail, and give some opinions and thoughts.

Minor problem:

There are many grammatical or format errors in the manuscript of the article, which need to be modified. Only a few are listed here.

1.      Some grammatical errors:the sentence” we demonstrated that miR-150 was induced by PRRSV via activating PKC/JNK/c-Jun pathway on lines 17-18” is not clear.

2.      Some format problems: The first occurrence of the abbreviation should show the full name. For example, PRRSV in the title and JAK/STAT1 in line 50, PEDV on line 82, qRT-PCR on line 107, GAPDH and U6 one line 116, Abs on line 125 and so on.

3.      On lines 72-73: peripheral blood mononuclear cells are often referred to as PBMC rather than BMo.

4.      On lines 81-82: Virus data in brackets should be after the virus strain and not at the end of the sentence.

5.      On lines 98-101,124-125,135: the source of antibody was not described.

Author Response

Dear Reviewer,

We appreciate the efforts of the reviewers for the careful review and the critical comments on our manuscript.

Thank you very much!

Sincerely,

Wen-hai Feng, DVM PhD

Reviewer 2 Report

The paper deals with a very interesting topic using a suitable and multidisciplinary approach. I have some concerns about the way data are provided.

Lines 59-68: the aims of the study should be clearly stated at this point, whereas Authors report a kind of summary of the paper.

Results: actually, each paragraph contains a mix of materials and methods, results and discussion. As an example, data reported at lines 203-223 have no link with materials and methods. Likewise, lines 170-175 should not be included in the results paragraphs.

As a consequence, materials and methods, as well as discussion, should be modified and implemented, also to reach a wider audience.

The English format should be carefully checked.

Author Response

(The authors gave the same response as above.)

Reviewer 3 Report

The work by Li et al., titled' Inducible miR-150 Inhibits PRRSV Replication by Targeting 2 Viral Genome and Suppressor of Cytokine Signaling 1' is overall an interesting study.

1. A general comment is to improve the figure quality and increase the size of individual panels (specifically Figure 3 and 6). Also, scale bars should be included in Figure 4.

2. Line 450 ' I think the authors mean ythat miR-150 is a negative regulator of PRRSV replication.

Author Response

Dear Reviewer,

We appreciate your efforts in the careful review and the critical comments on our manuscript.

Please see the attachment for the response.

Thank you very much!

Sincerely,

Wen-hai Feng, DVM PhD
